# FineScope: SAE-guided Data Selection Enables Domain-Specific LLM Pruning Fine-Tuning

## Abstract

Large Language Models (LLMs) are typically trained on diverse, general-purpose datasets, enabling broad generalization but incurring substantial computational costs. However, real-world applications often require efficient models tailored to narrow, domain-specific tasks. In such settings, large model capacity and generality are unnecessary, and traditional fine-tuning pipelines struggle under resource constraints. We introduce FineScope, a framework that addresses this challenge by tightly coupling domain-aware data selection with model pruning and fine-tuning. Starting from a small set of user-provided seed examples, FineScope trains sparse autoencoders (SAEs) on intermediate model activations to automatically extract semantically aligned examples from large unlabeled corpora. The curated dataset then guides structured pruning to preserve domain-relevant substructures and supports self-distillation fine-tuning to recover task-specific performance. Experiments across STEM, humanities, social sciences, math, and coding domains show that FineScope consistently outperforms baseline fine-tuning approaches while enabling up to $35\%$ parameter pruning. On math reasoning tasks, it achieves an average improvement of 11.50 points across pruned models. Code will be available.

## 1 Introduction

Large Language Models (LLMs) have demonstrated strong generalization across a wide range of tasks due to their training on broad and diverse datasets(RBC Borealis AI, 2024; Adler et al., 2024). However, most real-world applications run in resource-constrained environments (Kim et al., 2023; Dubey et al., 2024; Wang et al., 2024) and require models to perform well on a narrow set of domain-specific tasks. In these cases, the broad capabilities of large models are often unnecessary, and much of their computational cost is spent on supporting functions that are irrelevant to the target domain. This creates a growing need for models that are both efficient and specialized for specific applications. Despite significant progress in model compression techniques such as pruning and

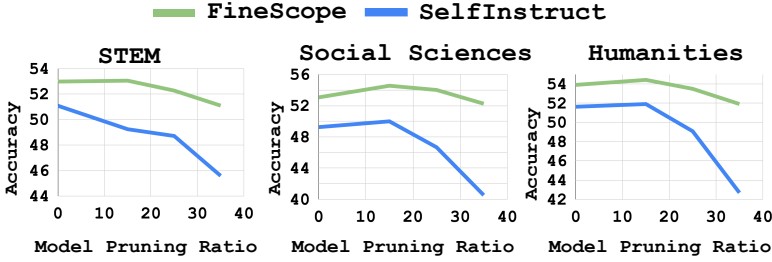

Figure 1: Overall accuracy (with different model pruning ratio) on the STEM, Social Sciences, and Humanities domains after fine-tuning with Self-Instruct dataset (Wang et al., 2022) and FineScope dataset. Despite its smaller size, FineScope sustains higher accuracy under aggressive pruning, underscoring its data efficiency and quality.

quantization, effective domain adaptation still depends heavily on the availability of high-quality, domain-specific data. In an ideal setting, a well-constructed dataset tailored to the target domain would be used for fine-tuning (Talmor et al., 2018; Clark et al., 2018; Gu et al., 2024). However, such datasets are rarely available in practice. Most existing approaches rely on general-purpose instruction datasets (Taori et al., 2023) or manually curated corpora, which can be noisy, expensive to build, or misaligned with the intended application. As a result, models are often fine-tuned with suboptimal data which hinders performance recovery after compression.

A key challenge in domain adaptation is the limited availability of high quality, domain aligned data for fine tuning. While much of the existing work focuses on model compression and adaptation techniques, these approaches often assume access to suitable data or treat data selection as a secondary concern. In practice, the effectiveness of compressed models depends heavily on the quality and relevance of the fine tuning dataset (Zhou et al., 2024). Without data that closely reflects the target domain, even the most advanced compression strategies are unlikely to maintain strong performance.

To address this challenge, we propose that data selection should be treated as a central part of the adaptation process. Inspired by findings that data quality can be more impactful than quantity (Zhou et al., 2024), we argue that a small, carefully chosen subset of relevant data can support strong performance, even in heavily compressed models. Rather than treating data curation and model optimization as separate steps, we design an approach that connects them, allowing the data to guide both pruning and fine-tuning within a unified framework.

We present FineScope, a framework that automates domain-specific data selection and integrates it with model pruning and fine-tuning. FineScope begins with a small set of user-provided seed examples and uses a Sparse Autoencoder (SAE), trained on intermediate activations of a pretrained LLM, to identify semantically relevant samples (Templeton et al., 2024; Kissane et al., 2024; Yan et al., 2024) from a large unlabeled corpus. These curated samples form a compact, high-quality dataset that reflects the target domain and is used to guide structured pruning. A modified self-distillation fine-tuning step then helps recover any task-relevant knowledge lost during compression.

FineScope enables the development of lightweight, domain-specialized models with minimal supervision and computational cost. As illustrated in Figure 1 the framework combines automatic data selection, structured pruning, and fine-tuning to produce compact models that retain domain-specific performance. Across a range of domains and tasks, our experiments show that FineScope outperforms standard fine-tuning pipelines and significantly improves the performance of pruned models. These results demonstrate that integrating targeted data selection with model adaptation is a powerful and practical strategy for domain-specific LLM deployment in resource-constrained environments. Our contributions are as follows:

- We propose FineScope, a unified framework that connects domain-specific data selection with model pruning and fine-tuning to support efficient adaptation of large language models.

- We introduce a novel use of Sparse Autoencoders trained on intermediate activations to identify semantically relevant data samples from large unlabeled corpora, starting from only a small seed set.

- We develop a modified self-distillation fine-tuning approach that helps pruned models regain domain-relevant behaviors using the curated dataset.

- We demonstrate that FineScope consistently improves performance over standard fine-tuning methods and enables effective domain adaptation, while significantly reducing model size.

## 2 RELATED WORK

*(1) Domain-Specific Language Models* Recent efforts have adapted large language models to specialized domains by training or fine-tuning them on domain-specific datasets. Examples include PharmaGPT (Chen et al., 2024) for biomedical applications, SaulLM (Colombo et al., 2024) for legal tasks, Shai (Guo et al., 2023) for asset management, BloombergGPT (Wu et al., 2023) for financial analysis, and MedPalm (Singhal et al., 2023) for medical question answering. Additional models

such as ClimateBERT (Webersinke et al., 2021), ChatLaw (Cui et al., 2023), and FinGPT (Yang et al., 2023) focus on areas such as climate science, legal reasoning, and financial modeling. While effective, these models typically rely on access to large, high-quality domain datasets and require full-scale retraining of billion-parameter models. Few methods address the challenges of adapting models efficiently, particularly in settings with limited compute and annotated data. In contrast, our work focuses on enabling domain adaptation by automatically selecting relevant data from large unlabeled corpora and compressing models to reduce computational requirements without sacrificing task performance.

*(2) Pruning* Pruning is a widely used technique for reducing the size and computational cost of language models by removing parameters that have minimal impact on performance. Motivated by the lottery ticket hypothesis (Frankle & Carbin, 2018), many methods aim to identify smaller subnetworks within large models that can be trained or fine-tuned to match the original model's accuracy. Structured pruning, in particular, removes entire architectural components such as attention heads or feedforward blocks, resulting in models that are both compact and efficient in hardware implementation.

Existing pruning methods, both structured and unstructured (Wang et al., 2019; Xia et al., 2022; Zafrir et al., 2021; Kurtic et al., 2022; Ashkboos et al., 2024; Xia et al., 2023), are typically designed to maintain general-purpose capabilities and are applied independently of the data used for downstream fine-tuning. As a result, they may not fully account for the specific needs of domain adaptation. In our approach, pruning is informed by data that is automatically selected for its relevance to a target domain. By integrating data selection into the pruning process, where we prune the model with respect to the domain-specific dataset, we aim to retain subnetworks that are more closely aligned with domain-specific behavior.

*(3) Neural Representation Alignment* While data-centric adaptation has gained increasing attention, most methods treat the model as fixed and focus solely on selecting or generating training examples. Instruction tuning and self-training approaches (Zhou et al., 2024; Taori et al., 2023; Wang et al., 2022) improve performance by using curated prompts or synthetic data but do not adapt the underlying model structure. Retrieval-based methods (Bricken et al., 2023; Gadre et al., 2023) select data based on surface-level similarity or metadata, which may not accurately reflect the model's internal understanding of domain relevance.

Recent studies suggest that intermediate model activations encode meaningful signals related to both task characteristics and input semantics (Templeton et al., 2024). However, this insight has rarely been applied to guide either data selection or model compression. Our work builds on this idea by using a Sparse Autoencoder trained on the top-$k$ intermediate activations of a pretrained model to estimate domain relevance. This allows us to identify training data that aligns with the model's internal representations and use it to inform both pruning and fine-tuning.

To the best of our knowledge, FineScope is the first framework to jointly leverage latent model representations for both dataset construction and structured pruning, enabling efficient domain adaptation under resource constraints.

## 3 FINESCOPE

We present FineScope, a two-stage framework for efficient domain adaptation of large language models. The first stage selects domain-relevant examples from a large unlabeled corpus using a sparse autoencoder trained on the model's internal activations. In the second stage, the selected data is used to guide structured pruning and fine-tuning through a modified self-distillation process. This approach enables the specialization of compact models that retain strong performance within the target domain.

### 3.1 SPARSE-AUTOENCODER-GUIDED DOMAIN-SPECIFIC DATASET CURATION

#### 3.1.1 TRAINING SAE

A Sparse Autoencoder (SAE) is a neural network designed to learn compressed representations of input data while enforcing sparsity constraints on the hidden units (EleutherAI, 2024). In our framework, SAEs serve as a mechanism for extracting domain-relevant features from pretrained LLM

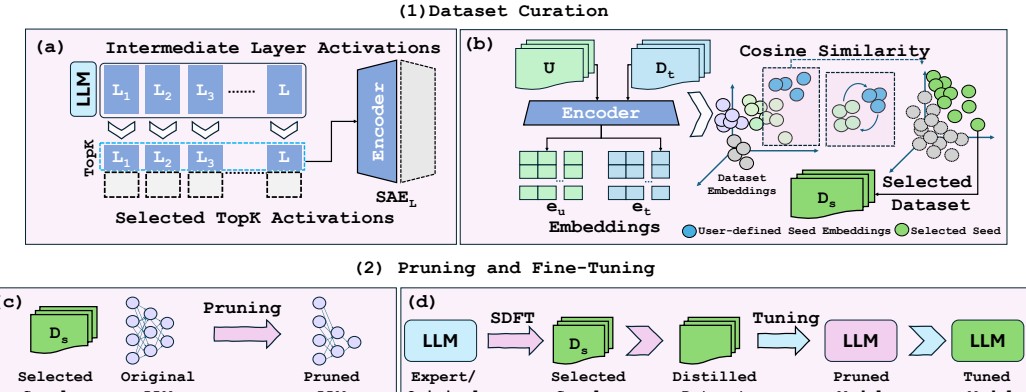

Figure 2: Overview of the FineScope : (1) *Dataset Curation*: (a) Sparse Autoencoder (SAE) is trained on the top-$K$ activations of a pretrained LLM and then used it to extract embedding from datasets,(b)Domain-specific dataset is curated by computing cosine similarity between target domain and the samples in the larger dataset. (2) *Pruning and Fine-Tuning*: (c) Structured pruning is done w.r.t. selected dataset; (d) Fine-tuned the pruned model using modified self distillation. Here, $U$ denotes the larger dataset, while $D_t$ represents the target domain dataset. The corresponding embeddings are denoted by $e_u$ and $e_t$ respectively.

activations to identify domain-relevant samples from a large corpus. Instead of operating directly on the raw model outputs, we train SAEs on activations from intermediate layers of the LLM, allowing us to capture a structured, low-dimensional representation of the underlying knowledge encoded in the model, as shown in Figure 2a. Additionally, since processing all activations are computationally infeasible for the large corpus, our SAEs are adapted to learn only the representative activations in a way that highlights the most significant neurons, improving interpretability while discarding less relevant signals.

In decoder-only transformer models such as GPT-2 and LLaMa, the activation flow can be formally defined as follows (Braun et al., 2024):

$$act^{(l)}(x) = f^{(l)}\left(act^{(l-1)}(x)\right), \text{for } l = 1, \ldots, L-1 \tag{1}$$

where $act^{(l)}(x)$ denotes the activations at layer $l$ given input $x$, and $f^{(l)}$ represents the transformation function at layer $l$, which typically includes multi-head self-attention, feed-forward operations, and residual connections. The final model output is computed as:

$$y = \text{softmax}\left(f^{(L)}\left(act^{(L-1)}(x)\right)\right). \tag{2}$$

To train the SAE, we extract activations $act^{(l)}(x)$ from a selected layer of the pretrained LLM and feed them into the encoder network:

$$\text{Enc}\left(act^{(l)}(x)\right) = \text{ReLU}\left(W_e act^{(l)}(x) + b_e\right). \tag{3}$$

The corresponding reconstruction from the decoder is given by:

$$\text{SAE}\left(act^{(l)}(x)\right) = Dec^{\top}\text{Enc}\left(act^{(l)}(x)\right) + b_{dec}. \tag{4}$$

Here, Enc represents the SAE's encoder, $W_e$ and $b_e$ are the encoder's weights and biases, and $Dec^{\top}$ and $b_{dec}$ correspond to the decoder parameters. The SAE is trained to minimize the reconstruction loss:

$$\mathcal{L}_{\text{SAE}} = \left\|\text{SAE}(act^{(l)}(x)) - act^{(l)}(x)\right\|_2^2 + \lambda\left\|\text{Enc}(act^{(l)}(x))\right\|_1. \tag{5}$$

where the second term enforces sparsity by penalizing the activation magnitudes of the encoder

### 3.1.2 TOP-K ACTIVATION SELECTION FOR EFFICIENT SAE TRAINING

Rather than training the SAE on full-layer activations, we apply a *Top-K filtering mechanism* to select the most important activations before feeding them into the encoder. Given dataset $D$ containing $m$ samples, $D = \{x_1, x_2, \ldots, x_m\}$, we compute the $K$ most significant activations based on the gradient magnitude:

$$\text{TopK}(act^{(l)}(x), K) = \{a_i \in act^{(l)}(x) \mid a_i \text{ is one of the } K \text{ largest } \left| \frac{\partial act_i^{(l)}(x)}{\partial x} \right| \}. \tag{6}$$

Using only the $TopK$ activations reduces the input dimensionality of the SAE, significantly lowering computational overhead and enabling efficient training on large-scale corpora. Additionally, filtering out noisy activations improves generalization by focusing the SAE on the most informative neurons, enhancing interpretability of extracted features aligned with domain-relevant information.

The encoder then operates on these filtered activations as:

$$\text{Enc}\left(\text{TopK}(act^{(l)}(x), K)\right) = \text{ReLU}\left(W_e \cdot \text{TopK}(act^{(l)}(x), K) + b_e\right). \tag{7}$$

For each layer $l$ in the LLM, we train a separate SAE, resulting in a total of $L$ SAEs. Once trained, these SAEs serve as *feature extractors for data curation*.

### 3.1.3 DATASET CURATION

After training, each SAE functions as a feature extractor to identify domain-relevant samples from a large unlabeled corpus. Starting with a small number of seed examples representative of the target domain (e.g., around ten samples), we aim to construct a curated subset $D_s \subseteq U$ from a broader, mixed-domain dataset $U$ by selecting examples that are most similar to the seed set in the learned embedding space. This selection process, which aligns samples in $U$ with the target domain using SAE-derived representations, is shown in Figure 2b.

Using the trained SAE, we compute embeddings for all samples in both $D_t$ and $U$. The embeddings for the target domain are given by $E_t = \{\text{SAE}(x) \mid x \in D_t\}$. Similarly, we compute embeddings for all samples in the larger dataset $U$, $E_U = \{\text{SAE}(x) \mid x \in U\}$.

To identify samples in $U$ that are most similar to $D_t$, we compute the cosine similarity between each embedding $e_u \in E_U$ and every embedding $e_t \in E_t$. Using this similarity measure, we select samples $x_u \in U$ whose embeddings fall within the Top-K highest cosine similarity scores when compared to the embeddings in $E_t$. Specifically, we define the selected dataset $D_s$ as:

$$D_s = \{x_u \in U \mid e_u \in \text{TopK}(\{\text{CosSim}(\text{SAE}(x_u), e_t) : e_t \in E_t\})\}. \tag{8}$$

where $\text{CosSim}(e_u, e_t)$ denotes the cosine similarity.

The final dataset $D_s$ contains samples from $U$ that are most semantically similar to the target domain $D_t$ based on the cosine similarity of their SAE embeddings. To ensure that $D_s$ remains a high-quality domain-specific dataset. In our evaluation, the value of $K$ across all SAEs is set to 100 for consistency in selection.

## 3.2 PRUNING AND FINE-TUNING WITH MODIFIED SDFT

### 3.2.1 PRUNING

To enhance model efficiency while retaining domain-specific knowledge, we apply structured pruning using LLM-Pruner (Ma et al., 2023). Rather than relying on general-purpose criteria, we guide the pruning process with a domain-specific dataset $D_s$ (as shown in Figure 2c), so that only components relevant and active to the target domain are preserved. Guided by the domain-specific dataset $D_s$, the contribution of each model block is estimated using gradient-based attribution. Specifically, we compute an importance score for each block based on the first-order gradient of the task-specific loss with respect to $D_s$. This allows the pruning process to preserve components that are most critical for performance in the target domain.

We apply block-wise structured pruning using LLM-Pruner (Ma et al., 2023), which removes redundant components of the model and significantly reduces inference cost. The original model, denoted by $\mathcal{M}$, is pruned with respect to the domain-specific dataset $D_s$ and a pruning ratio $r$, resulting in a compressed model $\mathcal{M}_r$:

$$\mathcal{M}_r = \text{LLMPrune}(\mathcal{M}, D_s, r) \tag{9}$$

Here, $r$ serves as a hyperparameter that controls the trade-off between model compactness and domain-specific performance.

### 3.2.2 Fine-Tuning

Self Distillation Finetuning (SDFT) (Yang et al., 2024) is a technique in which a distilled dataset is generated to match the output distribution of the original model, leading to improved generalization. While SDFT is commonly used for knowledge transfer, its role in FineScope extends beyond dataset refinement. We adapt SDFT to address the loss of domain-specific representations caused by pruning. High-confidence predictions from a state-of-the-art teacher model are used to generate a distilled dataset, which is then used to fine-tune the pruned student. This teacher-guided distillation helps restore lost knowledge and improves performance. A high level overview of the process is illustrated in Figure 2d. This approach reduces overfitting to the small curated dataset and compensates for information removed during pruning, even when the teacher and student differ in size or architecture.

In our framework, we adapt SDFT to further refine the pruned model. We generate a distilled dataset using either the original model or a pretrained state-of-the-art teacher, and use it to fine-tune the pruned model. This process helps preserve domain-specific knowledge while improving generalization.

In the original SDFT formulation (Yang et al., 2024), given an input $x$, context $c^t$, and output $y^t$ from the teacher model, the distilled output $y'$ is sampled as:

$$y' \sim f(y \mid c^t, x^t, y^t). \tag{10}$$

We adapt this procedure in FineScope by modifying the fine-tuning objective for the pruned model. The resulting self-distillation loss is defined as:

$$L_{\text{msdft}} = -\log f_p(y' \mid c^t, y^t), \tag{11}$$

where $f_p$ denotes the pruned model and $L_{\text{msdft}}$ represents the self-distillation loss.

## 4 Evaluation

*(1) Models.* We evaluate our method using three models: Vicuna-7B (Zheng et al., 2023), a fine-tuned version of LLaMa 2 (Touvron et al., 2023); MathCoder-CL-7B (Wang et al., 2023), a CodeLlama (Roziere et al., 2023) variant; and LLaMa 3.1-8B (Dubey et al., 2024).

*(2) Baselines.* We compare FineScope against six baseline settings: (a) fine tuning the pruned model using randomly selected data of the same size, (b) fine tuning with the full dataset containing mixed domains, (c) fine tuning with Alpaca data using FineScope's pruning strategy, (d) evaluating pretrained models without any fine tuning, (e) evaluating pretrained models fine tuned with FineScope-curated data, and (f) comparisons against GPT-3 (6.7B and 175B)(Brown et al., 2020) and OLMO-7B(Groeneveld et al., 2024) (Table 1), along with GPT-3 (13B and 175B) (Table 2)

*(3) Tuning Tasks.* We assess FineScope on three main tasks:

*(a) Domain Specific Tuning.* We prune models using domain specific datasets curated with our SAE-guided framework. The SAEs are trained on the RedPajama dataset (Computer, 2023), which includes content from CommonCrawl, C4, GitHub, Wikipedia, Books3, ArXiv, and StackExchange, providing broad domain coverage. Using these SAEs, we curate domain specific subsets from OpenInstruct (hakurei, 2023), which aggregates instruction datasets such as Alpaca (Lan, 2019), Self Instruct (Wang et al., 2022), GPT 4 Instruct, Roleplay (Teknium, 2023), Code Alpaca (Chaudhary, 2023), and Dolly (Ouyang et al., 2022). Based on user provided seed samples, we extract curated datasets for STEM (2,100 samples), Social Sciences (2,401 samples), and Humanities (2,374 samples), selecting the most frequently chosen samples across all trained SAEs.

*(b) Subdomain Specific Tuning (Math).* To assess FineScope's effectiveness at a more granular level, we evaluate fine tuning on mathematical subdomains. SAEs are trained on the MetaMath dataset (Yu et al., 2023), and used to curate subsets from the Math dataset (Hendrycks et al., 2021). In this setting, subdomains are merged into a unified pool for curation. From this pool, we extract the Pre Algebra, Algebra, and Counting and Probability subsets. Models are fine tuned using Open-Math2 (Toshniwal et al., 2024), and Notus 7B (Tunstall et al., 2023) serves as the Alpaca tuned baseline. Evaluations are conducted separately on each subdomain test set (Table 2).

*(c) Coding Specific Tuning.* Following the same SAE guided curation approach, we construct a code focused dataset from OpenInstruct, resulting in 1,200 examples. To evaluate model performance on code generation, we fine tune on this curated dataset and assess results using the HumanEval (Chen et al., 2021) and MBPP (Austin et al., 2021) benchmarks (Table 3).

*(4) Implementation Details SAE training:* We ran SAE training (EleutherAI, 2024) using the AdamW optimizer with a learning rate of 1e-5, a batch size of 8, and a Top-K value of 128. We used GPT-4 (Achiam et al., 2023) to generate the 10 user-defined seeds. *Finetuning:* We fine-tuned LMs (LORA-fine tuning (Hu et al., 2021)) using the same AdamW optimizer at a 5e-5 learning rate, a batch size of 128, lora rank of 32, and a 256 cut-off length.

## 4.1 EXPERIMENTAL RESULTS

Table 1: Performance comparison of FineScope-tuned models versus baselines across STEM, Social Sciences (Social Sci.), and Humanities (Hum.) domains.

| Model | Pruned | Tuned | Dataset | STEM | Social Sci. | Hum. |
|---|---|---|---|---|---|---|
| | × | × | – | 33.10 | 40.23 | 43.69 |
| | ✓ | × | – | 17.17 | 20.11 | 20.80 |
| Vicuna (Zheng et al., 2023) | ✓ | × | Random | 18.52 | 21.29 | 20.21 |
| | ✓ | × | Full-OI | 29.09 | 35.43 | 36.19 |
| | ✓ | × | Alpaca | 30.61 | 35.44 | 36.11 |
| | × | ✓ | FineScope | 33.32 | 40.21 | 42.43 |
| | ✓ | ✓ | FineScope | **31.12** | **36.23** | **36.55** |
| | × | × | – | 31.14 | 11.11 | 9.22 |
| | ✓ | × | – | 13.32 | 8.02 | 3.67 |
| MathCoder-CL Wang et al. (2023) | ✓ | × | Random | 12.94 | 7.53 | 4.59 |
| | ✓ | × | Full-OI | 23.91 | 12.81 | 12.67 |
| | ✓ | × | Alpaca | 25.14 | 13.11 | 12.33 |
| | × | ✓ | FineScope | 34.96 | 32.91 | 31.66 |
| | ✓ | ✓ | FineScope | **25.89** | **13.81** | **13.68** |
| | × | × | – | 48.01 | 49.61 | 49.32 |
| | ✓ | × | – | 30.59 | 31.33 | 33.62 |
| LLaMa3.1 (Dubey et al., 2024) | ✓ | × | Random | 29.04 | 30.93 | 33.71 |
| | ✓ | × | Full-OI | 39.32 | 39.91 | 40.93 |
| | ✓ | × | Alpaca | 38.22 | 40.19 | 39.79 |
| | × | ✓ | FineScope | 48.84 | 51.66 | 51.45 |
| | ✓ | ✓ | FineScope | **40.55** | **41.07** | **41.19** |
| GPT-3 (6.7B) | × | × | – | 35.10 | 49.20 | 42.10 |
| OLMO | × | × | – | 22.19 | 31.01 | 30.26 |
| GPT-3 (175B) | × | × | – | 36.70 | 50.40 | 40.80 |

Table 1 presents evaluation results on the MMLU dataset across three domains. Models adapted using FineScope, through pruning and fine tuning with SAE-curated domain specific data, achieve average performance gains of 3.8% over Alpaca tuning and 4.45% over OpenInstruct (Full OI) across all domains and model types. Among the three evaluated models, MathCoder CL shows the most significant improvement, with gains of 8.28% in STEM, 7.8% in Social Sciences, and 7.9% in Humanities. These results indicate that SAE-guided data selection not only improves domain adaptation but also enables pruned models to recover performance that would otherwise be lost under aggressive compression. Despite using fewer data points, our method outperforms OpenInstruct, underscoring the importance of data quality and domain alignment over quantity alone.

Pruning without domain guidance results in substantial performance degradation, reaching up to 50.17% on average for Vicuna across all domains. LLaMa 3.1 shows the smallest drop, likely due to its more balanced initial performance and the ability of domain focused pruning to retain essential parameters. Compared to GPT-3 (6.7B and 175B) and OLMO 7B, our pruned models,

with approximately 30% fewer parameters, outperform in most settings. GPT-3 achieves stronger results in Social Sciences, and the 175B variant exceeds our models in Humanities, but FineScope tuned models consistently outperform OLMO 7B across all domains.

Table 2: Performance comparison of FineScope-tuned models versus baselines across Pre-algebra (Pre-alg.), Algebra (Alg.), and Counting and Probability (Count.&Prob.) domains.

| Model | Pruned | Tuned | Dataset | Pre-alg. | Alg. | Count.&Prob. |
|---|---|---|---|---|---|---|
| | ✗ | ✗ | – | 14.31 | 10.17 | 8.11 |
| | ✓ | ✗ | – | 0.11 | 0.00 | 0.00 |
| Vicuna (Zheng et al., 2023) | ✓ | ✗ | Random | 0.00 | 0.00 | 0.00 |
| | ✓ | ✗ | Full-Math | 12.73 | 8.91 | 5.48 |
| | ✓ | ✓ | Alpaca | 5.56 | 0.30 | 0.21 |
| | ✗ | ✓ | FineScope | 15.46 | 13.33 | 10.43 |
| | ✓ | ✓ | FineScope | **12.91** | **10.12** | **7.01** |
| | ✗ | ✗ | – | 11.60 | 16.77 | 13.38 |
| | ✓ | ✗ | – | 0.59 | 2.33 | 0.29 |
| MathCoder-CL (Wang et al., 2023) | ✓ | ✗ | Random | 0.00 | 0.00 | 0.00 |
| | ✓ | ✗ | Full-Math | 9.01 | 12.72 | 10.05 |
| | ✓ | ✓ | Alpaca | 1.29 | 6.94 | 3.33 |
| | ✗ | ✓ | FineScope | 14.73 | 17.75 | 15.43 |
| | ✓ | ✓ | FineScope | **10.54** | **15.51** | **11.64** |
| | ✗ | ✗ | – | 32.77 | 29.87 | 20.35 |
| | ✓ | ✗ | – | 11.41 | 7.99 | 5.01 |
| LLaMa3.1 (Dubey et al., 2024) | ✓ | ✗ | Random | 7.04 | 8.01 | 6.93 |
| | ✓ | ✗ | Full-Math | 30.72 | 31.67 | 18.34 |
| | ✓ | ✓ | Alpaca | 9.23 | 5.56 | 9.10 |
| | ✗ | ✓ | FineScope | 34.46 | 31.85 | 23.18 |
| | ✓ | ✓ | FineScope | **30.83** | **32.21** | **19.34** |
| GPT-3 (13B) (Brown et al., 2020) | ✗ | ✗ | – | 6.80 | 5.30 | 4.50 |
| GPT-3 (175B) (Brown et al., 2020) | ✗ | ✗ | – | 7.70 | 6.00 | 4.70 |

Table 2 highlights significant performance improvements in math domains using our domain-specific tuning: Vicuna (+7.01), MathCoder-CL (+7.71), and LLaMa 3.1 (+18.45) versus Alpaca-tuned baselines. A similar trend is observed when finetuned with Math's full corpus (e.g.,+1.97 average performance gain for MathCoder-CL when compared with Full-Math corpus). However, pruning severely degrades Vicuna and MathCoder-CL's performance and Alpaca's general-purpose instructions fail to restore performance due to a lack of semantic focus. Compared to GPT models, our tuned models achieve competitive performance, with differences likely due to GPT's imbalanced training data limiting generalization. Despite reducing model size to approximately 71% of the original, FineScope is able to restore performance by fine tuning on a semantically focused dataset.

As shown in Table 3, our domain specific tuning dataset, FineScope, substantially improves coding performance on the HumanEval and MBPP benchmarks, especially after model pruning. When ap-

Table 3: Performance comparison of FineScope-tuned models versus baselines across MBPP and HumanEval coding datasets.

| Model | Pruned | Tuned | Dataset | HumanEval | MBPP |
|---|---|---|---|---|---|
| | ✗ | ✗ | – | 0.14 | 0.03 |
| | ✓ | ✗ | – | 0.04 | 0.00 |
| Vicuna | ✓ | ✗ | Random | 0.03 | 0.00 |
| | ✓ | ✗ | Full-OI | 0.09 | 0.05 |
| | ✓ | ✓ | Alpaca | 0.07 | 0.00 |
| | ✗ | ✓ | FineScope | 0.21 | 0.13 |
| | ✓ | ✓ | FineScope | **0.13** | **0.10** |
| | ✗ | ✗ | – | 0.03 | 0.01 |
| | ✓ | ✗ | – | 0.00 | 0.00 |
| MathCoder-CL | ✓ | ✗ | Random | 0.00 | 0.00 |
| | ✓ | ✗ | Full-OI | 0.10 | 0.09 |
| | ✓ | ✓ | Alpaca | 0.08 | 0.05 |
| | ✗ | ✓ | FineScope | 0.20 | 0.14 |
| | ✓ | ✓ | FineScope | **0.11** | **0.10** |
| | ✗ | ✗ | – | 0.50 | 0.46 |
| | ✓ | ✗ | – | 0.26 | 0.13 |
| LLaMa3.1 | ✓ | ✗ | Random | 0.20 | 0.09 |
| | ✓ | ✗ | Full-OI | 0.30 | 0.29 |
| | ✓ | ✓ | Alpaca | 0.25 | 0.13 |
| | ✗ | ✓ | FineScope | 0.55 | 0.48 |
| | ✓ | ✓ | FineScope | **0.49** | **0.43** |

plied to pruned models, FineScope yields coding gains of +0.08 for Vicuna, +0.04 for MathCoder CL, and +0.27 for LLaMa 3.1 8B compared to tuning with the full Alpaca dataset. In all three models, pruning alone leads to a significant decline in code generation performance, and tuning on the Alpaca dataset fails to recover the loss.In comparison to OpenInstruct's full corpus (Full OI), FineScope delivers highly competitive and often superior results. For instance, the pruned LLaMa 3.1 model fine tuned with FineScope achieves scores of 0.49 on HumanEval and 0.43 on MBPP, outperforming the same model tuned with Full OI, which reaches 0.30 and 0.29, respectively. These

Table 4: Effect of Modified SDFT (MSDFT) and domain-specific pruning methods across domains.

| Domain | MSDFT | | Pruning | |
|--------|-------|------|----------|-----------|
| | W/O | W/ | FineScope | Bookcorpus |
| STEM | 30.54 | **31.12** | **31.12** | 28.64 |
| Social Sciences | 34.25 | **36.23** | **36.23** | 33.24 |
| Humanities | 33.15 | **36.55** | **36.55** | 31.03 |

results demonstrate that FineScope is more effective at producing specialized, high performing models than simply relying on a large general purpose corpus for tuning and pruning guidance.

## 4.2 EFFECT OF MSDFT AND PRUNING DATASET

*Modified SDFT* Table 4 shows a consistent improvement in performance when modified SDFT is applied, compared to standard fine tuning. Across all three domains, models fine tuned with SDFT outperform their counterparts. For example, we observe performance gains of 1.9% in STEM, 5.8% in Social Sciences, and 10.26% in Humanities. This consistent improvement across diverse domains highlights the effectiveness of a distillation based approach for enhancing model performance in domain specific adaptation. *Pruning Dataset* Table 4 highlights pruning with the FineScope domain aligned dataset, followed by fine tuning on the same domain data, results in an average accuracy gain of 11.8% compared to pruning with a general purpose corpus (Kobayashi, 2018). This demonstrates that incorporating domain specific samples during the pruning stage helps retain critical representations that are often lost when using generic data, leading to consistently higher performance across STEM, Social Science, and Humanities benchmarks.

## 4.3 EFFECT OF TOPK ON COMPUTATION

The choice of the hyperparameter $K$ for our Sparse Autoencoders (SAEs) reflects a trade-off between training efficiency and performance across domains. As shown in Figure 3, increasing $K$ improves average accuracy on STEM, Social Sciences, and Humanities tasks but also leads to longer training times across all transformer blocks of the Vicuna 7B model. We find that setting $K$=128 provides a favorable balance, yielding strong average accuracy while keeping training time tractable. Further increasing $K$ to 256 offers only marginal accuracy gains at the cost of significantly higher computational overhead.

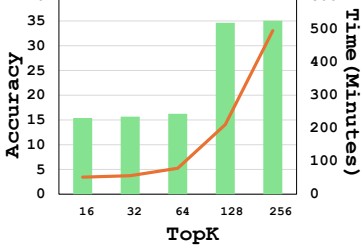

Figure 3: Impact of varying TopK on SAE's average reconstruction loss, average accuracy and training time for all transformer blocks.

## 5 CONCLUSION

We presented FineScope, a unified framework that enables efficient domain adaptation of LLMs by integrating sparse autoencoder-guided data selection with structured pruning and fine-tuning. By identifying semantically relevant samples from large unlabeled corpora, FineScope constructs compact, high-quality datasets that guide both compression and adaptation. Across diverse domains, FineScope consistently improves performance while significantly reducing model size, demonstrating that targeted data selection is crucial for effective LLM deployment in resource-constrained settings.

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

## A APPENDIX FOR FINESCOPE

### A.1 PRUNING RATIO BOUNDARY

To examine the limits of model compression, we compared performance under different pruning ratios using our FineScope pipeline and the larger SelfInstruct dataset on the LLaMA 3.1 8B model. As shown in Figure 4, FineScope demonstrates greater resilience to aggressive pruning. Although both methods show a decline in accuracy as pruning increases, FineScope consistently outperforms SelfInstruct across all settings. Notably, SelfInstruct begins to degrade at just 25% pruning, while FineScope maintains stable accuracy up to 35%. This indicates that FineScope enables higher pruning tolerance without compromising task performance.

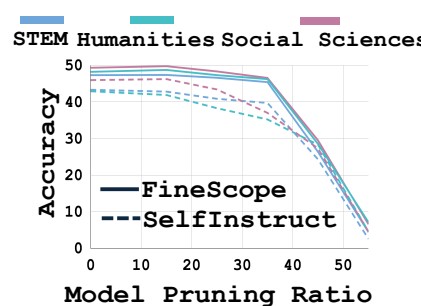

Figure 4: Effect of model pruning ratio on accuracy.

### A.2 COMPARISON WITH SYNTHETIC DATASET

In Figrue 5 we evaluate our curated FineScope dataset (2.1K samples) against the publicly available synthetic STEM-Saraswati dataset (Tiwari, 2024), generated using GPT-4 (Achiam et al., 2023), as well as general-purpose finetuning datasets such as Alpaca and OpenInstruct. scope achieves comparable accuracy to STEM-Saraswati on STEM-specific tasks, demonstrating the high quality of our curated dataset. Moreover, when contrasted with general-purpose datasets, both STEM-Saraswati and FineScope achieve substantially higher performance, highlighting the critical role of high-quality, domain-specific data in enhancing model capabilities within specialized scientific domains.

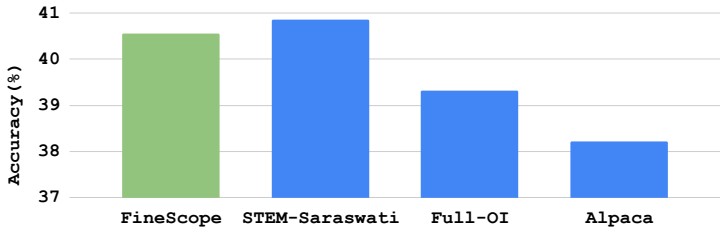

Figure 5: Performance comparison with synthetic STEM dataset.

### A.3 EFFECT OF SAE-EMBEDDING

As shown in Table 5, the method of data curation has a significant impact on model performance after fine-tuning. The results consistently demonstrate that curating the fine-tuning dataset using our interpretable SAE-embeddings leads to substantially better outcomes than using standard raw embeddings. To analyze the effect of the embedding type, we compare the performance of original and pruned models fine-tuned on datasets curated by each method. Using SAE-embeddings yields superior results across all models and domains. For example, with Vicuna the average performance gain after finetuning the original model and pruned model is + 5.08 and + 2.06 respectively across all domains. This indicates that the features captured by the SAEs are more relevant and lead to a higher quality fine-tuning dataset.

Table 5: Performance comparison of SAE-Embedding dataset selection versus BERT-Embedding dataset selection. MC-CL: MathCoder-CL.

| Model | Type | Dataset | STEM | Social Sci. | Hum. |
|-------|------|---------|------|-------------|------|
| Vicuna | Full | BERT-Embedding | 31.17 | 35.04 | 34.49 |
| | Pruned | BERT-Embedding | 29.01 | 34.23 | 34.48 |
| | Full | SAE-Embedding | **33.32** | **40.21** | **42.43** |
| | Pruned | SAE-Embedding | **31.12** | **36.23** | **36.55** |
| MC-CL | Full | BERT-Embedding | 30.45 | 29.14 | 29.01 |
| | Pruned | BERT-Embedding | 24.91 | 12.73 | 12.53 |
| | Full | SAE-Embedding | **34.96** | **32.91** | **31.66** |
| | Pruned | SAE-Embedding | **25.89** | **13.81** | **13.68** |
| LLaMa3.1 | Full | BERT-Embedding | 46.19 | 49.22 | 50.94 |
| | Pruned | BERT-Embedding | 37.02 | 39.32 | 39.00 |
| | Full | SAE-Embedding | **48.84** | **51.66** | **51.45** |
| | Pruned | SAE-Embedding | **39.91** | **41.07** | **41.19** |

## B  SEED SELECTION:

We visualize the seed selection process using cluster analysis, as shown in **Figure 6**. The interpretation of features varies across initial, middle, and final layers, reflecting how representations evolve within the model. The results suggest that our curated dataset is alinged with the clustered virtual domain target domains across different layer representations. Lower layers tend to capture broad, generalized features, often encoding syntactic structures or common linguistic patterns. In contrast, deeper layers focus on increasingly abstract and domain-specific attributes, leading to more compact and semantically meaningful clusters.

This progressive refinement suggests that domain-relevant information emerges more distinctly in later layers, where feature representations become more specialized. Our method leverages this hierarchical structure, selecting seeds from layers that balance generalization and domain specificity. By utilizing SAE-based representations instead of raw embeddings, we ensure that the seed selection process is more interpretable and aligned with high-level domain knowledge, rather than being influenced by superficial token-level similarities. This reinforces the effectiveness of our approach in identifying the most informative seed samples for dataset curation.

### B.1  EXAMPLE SEEDS

Figures 7, 8, and 9 present the user-defined seed samples for STEM, Social sciences and Humanities respectively. Based on these seeds, we extracted domain-specific data points from the larger dataset.

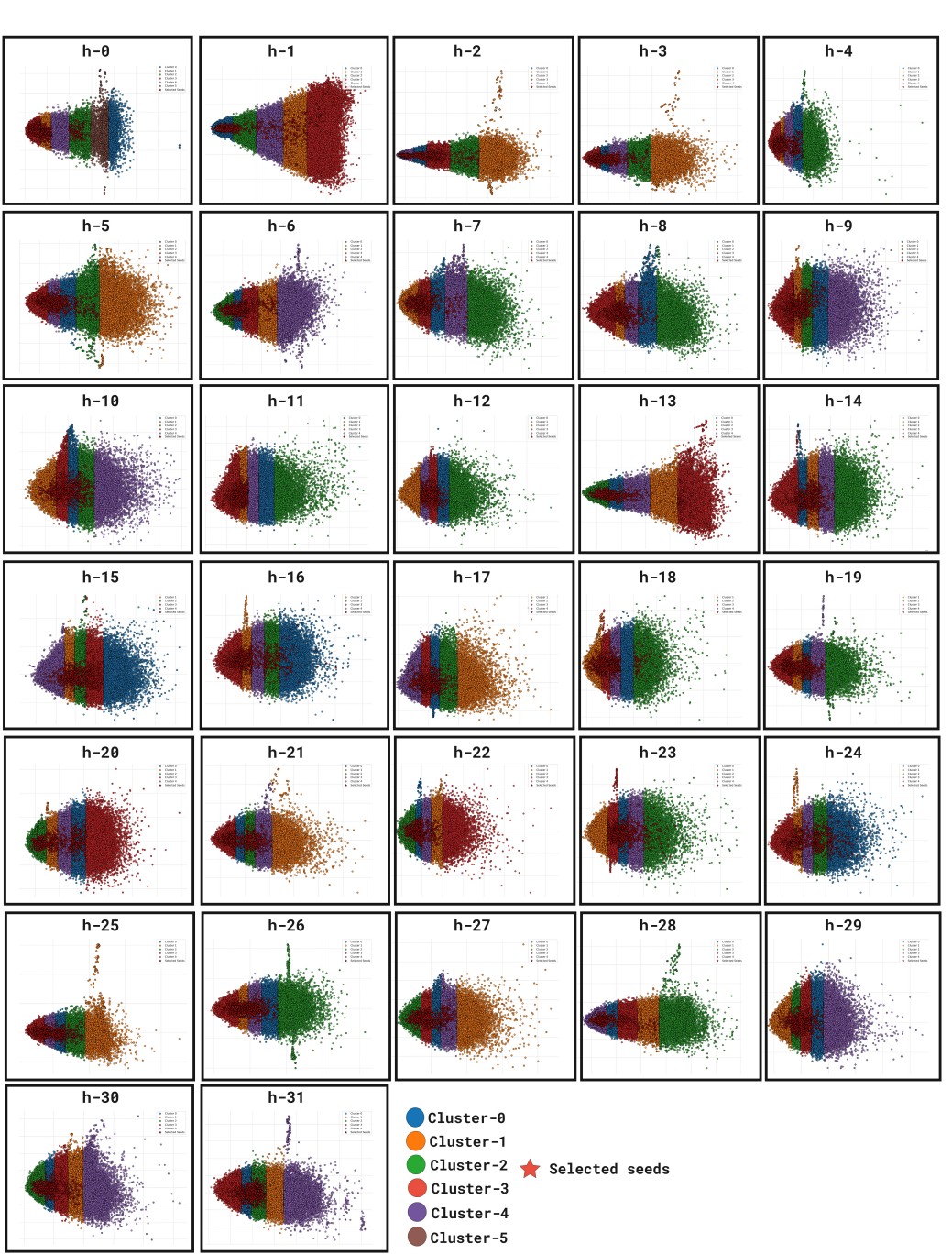

Figure 6: Cluster visualization for seed selection for dataset curation (for STEM)

**STEM**

Which type of reinforcement learning strategy involves explicitly learning a model of the environment?

Let L = Q(sqrt(3), sqrt(5)). What is the degree [L : Q]?

In a second-order linear control system, what does the damping ratio ζ determine?

The primary function of the mitochondria in eukaryotic cells is to:

Powerhouse of the Cell is :

In machine learning, what is the bias-variance tradeoff?

A NAND gate has two inputs. For how many input combinations does it output 1?

Let f(x) = x^4 - 4x^2 + 2. What is the degree of its splitting field over Q?

What does an increase in refractive index imply about the speed of light in a medium?

A computer algorithm has time complexity O(n log n). What is the best classification for this algorithm?

Figure 7: User defined seed sample of STEM target domain

**Social Sciences**

In a linear regression model, what does multicollinearity among explanatory variables cause?

Which of the following actions is most likely to reduce inflation in the short run?

What does Emile Durkheim mean by "anomie"?

What is the primary function of the legislative branch in the U.S. government?

Which geographic feature forms the natural border between France and Spain?

What does the law of diminishing marginal utility state?

The sexual response cycle as described by Masters and Johnson includes which of the following stages?

Which doctrine stated that the U.S. would provide military aid to countries resisting communism after World War II?

In cognitive-behavioral therapy, what is cognitive restructuring primarily used for?

What part of the brain is primarily associated with emotion processing?

Figure 8: User defined seed sample of Social sciences target domain

**Humanities**

**Which of the following actions is most likely to reduce inflation in the short run?**

**What was the primary cause of the decline of the Roman Republic?**

**In Shakespeare's "Macbeth", what prophecy do the witches give Macbeth?**

**The Enlightenment emphasized which of the following ideals?**

**Which language family does Finnish belong to?**

**What is the primary function of the legislative branch in the U.S. government?**

**Who wrote "The Second Sex", a foundational text in feminist philosophy?**

**What is the main theme of George Orwell's "1984"?**

**The Treaty of Versailles ended which major conflict?**

**Structuralism in literary theory is primarily concerned with:**

Figure 9: User defined seed sample of Humanities target domain

