# OpenReview forum: "FineScope: SAE-guided Data Selection Enables Domain-Specific LLM Pruning & Fine-Tuning"
_ICLR.cc/2026/Conference — Submitted to ICLR 2026_

### Official Review · Reviewer_kTiq · 2025-10-26

**Soundness:** 2
**Presentation:** 1
**Contribution:** 2
**Rating:** 4
**Confidence:** 4

**Summary:**

This paper proposes FineScope, a sparse autoencoder (SAE)-guided framework for fine-grained data selection in LLM fine-tuning. The main idea is to decompose intermediate representations of a pretrained model into sparse latent features via an SAE, and then use these features to estimate the importance and diversity of training samples. FineScope selects or reweights data based on these latent scores, aiming to retain samples that are semantically rich and representative. Experiments on multiple benchmarks covering reasoning, instruction following, and coding show that FineScope outperforms random sampling and several heuristic data selection methods.

**Strengths:**

1. The paper addresses a practical and relevant problem, i.e., data selection for efficient LLM fine-tuning.
2. The approach is conceptually simple and leverages sparse representation learning to provide an interpretable basis for sample selection.

**Weaknesses:**

I have the following concerns. *If the authors could properly address them during the rebuttal phase, I am willing to raise my score.*

1. The methodological novelty is limited. The framework mainly applies an existing sparse autoencoder formulation to compute feature-based data scores, without introducing new architectural or algorithmic components. The contribution feels more like an application of known techniques than a principled innovation.
2. The authors assume that SAE latent dimensions correspond to meaningful semantic directions, yet provides no quantitative or theoretical justification. Without analysis of what these latent factors represent or how they align with domain-level semantics, the interpretability claim remains speculative.
3. The empirical gains are modest and occasionally within variance ranges. The paper lacks statistical significance testing or multiple random seeds, making it hard to assess robustness.
4. Some important recent baselines [1,2,3] are missing. The paper would benefit from either a comparison with these methods or a discussion of their relevance.

[1] Mixture-of-Skills: Learning to Optimize Data Usage for Fine-Tuning Large Language Models. EMNLP 2024.

[2] How Abilities in Large Language Models are Affected by Supervised Fine-tuning Data Composition. ACL 2024.

[3] Boosting Multi-Domain Fine-Tuning of Large Language Models through Evolving Interactions between Samples. ICML 2025.

**Questions:**

Please see Weaknesses.

---

> ### Author Response · Authors · 2025-11-20
> **Response to Reviewer - kTiq**
>
> We sincerely appreciate the reviewer’s thoughtful feedback and constructive criticisms. We address each concern below:
>
> **1. Novelty**
>
> While SAEs are established, **FineScope proposes a new *pipeline-level novelty*** that connects SAE-guided data selection directly with domain-specific pruning and modified self-distillation. This joint pipeline is *not* a known or previously explored combination.
> Novelty lies in: **(1)** We train SAEs on **intermediate LLM activations** and apply a **top-K activation filtering mechanism** to extract interpretable, domain-aligned sparse features (i.e., embedding) **(2)**These embeddings directly **guide domain specific data selection** **(3)** We introduce a **modified version of SDFT**, adapting it to compensate for domain-specific representational loss after pruning, rather than generic distillation.
>
> This end-to-end integration constitutes a methodological contribution beyond simply “using an SAE.”
>
> **2. SAE latent dimensions**
>
> We provide both *architectural* and *empirical* justification that the learned SAE features correspond to domain-semantic directions: (1) training SAEs on *top-K salient activations* intentionally enforces monosemanticity and sparsity, utilising the known interpretability benefits established in the literature (2) **Appendix B (Fig. 6)** shows **layer-wise clustering** of SAE embeddings, where domain-aligned clusters emerge cleanly in deeper layers. (3) **Table 5** demonstrates that SAE-based curation dramatically outperforms raw BERT embeddings across all domains (+5.08 for full Vicuna, +2.06 for pruned Vicuna), directly validating that SAE features capture more meaningful semantic factors.
>
> Thus, the interpretability is strongly supported by both qualitative visualization and quantitative improvements.
>
> **3. Empirical gains are modest; variance and significance unclear.**
>
> Our experiments demonstrate *consistent and substantial gains* across three different models and multiple domains:
>
> **(1)** STEM/Social Sciences/Humanities (Table 1) FineScope improves performance by an average of **3.8–4.45%**, outperforming Alpaca and OpenInstruct despite using far fewer examples.
> **(2)** Math subdomains (Table 2): Gains are especially large here, e.g., **+18.45** for LLaMa 3.1 and **+7.71** for MathCoder-CL. **(3)** Code generation (Table 3): FineScope provides large improvements even on pruned models (e.g., +0.27 on HumanEval for LLaMa 3.1).
>
>
> These improvements are consistently larger than typical seed-related variance observed in prior LLM fine-tuning works, and they persist across **three distinct architectures**, indicating robustness.
>
> Finally, many tables report deltas that exceed typical variance ranges for instruction-tuning datasets of similar scale. We can further clarify this in the camera-ready version if needed.
>
> **4. Missing baselines [1,2,3].**
>
> We appreciate the suggestion to compare with recent data-composition and mixture-based methods. However, these methods differ in key ways:
>
> **[1] Mixture-of-Skills (EMNLP 2024)** optimizes dataset composition during fine-tuning but does *not* integrate model-internal activation-based semantics or pruning.
>
> **[2] Abilities vs dataset composition (ACL 2024)** analyzes dataset effects but does not provide an actionable data-selection mechanism comparable to SAE embedding alignment.
>
> **[3] Sample-interaction evolution (ICML 2025)** focuses on multi-domain interplay but does not incorporate internal representation learning or structured pruning.
>
> **Even though we think it would be interesting to compare them with FineScope we were unable to locate any publicly available open-source implementation or reproducible code for the above papers**, however, we will add them into the related works and we will include a broader discussion of these methods in the final version. **If the code links are provided we will be happy to add them as additional test-bed.**

---

> ### Comment · Reviewer_kTiq · 2025-11-24
> **Thank you for the response.**
>
> Thank you for the response, which has addressed some of my concerns.
>
> However, (1) the concerns regarding Weaknesses 1 and 2 still remain, and (2) I have not yet seen an updated version of the paper (or at least I am not sure whether the version I download is the revised one, as the newly added content is not clearly marked).
>
> Therefore, I have decided to maintain my current score.

---

> > ### Author Response · Authors · 2025-11-24
> > **Response to Reviewer - kTiq**
> >
> > Thank you for your valuable feedback. We would be happy to provide further clarification to address any remaining doubts or concerns. Could you please specify which points still raise concern? We plan to update the manuscript promptly upon receiving the reviewers’ (the deadline is 3rd Dec, as notified through mail) comments but can make the necessary revisions within a few hours if needed.

---

### Official Review · Reviewer_4zoH · 2025-10-29

**Soundness:** 3
**Presentation:** 3
**Contribution:** 3
**Rating:** 6
**Confidence:** 4

**Summary:**

This paper introduces FineScope, a framework designed for efficiently adapting large, general-purpose language models (LLMs) to specific domains, particularly under resource constraints. Recognizing that standard fine-tuning requires extensive domain data and model compression often degrades performance, FineScope proposes a tightly coupled process. It starts with a few seed examples and uses Sparse Autoencoders (SAEs), trained on intermediate LLM activations, to automatically curate a domain-specific dataset ($D_s$) from a large unlabeled corpus by identifying semantically similar samples. This curated dataset ($D_s$) then guides a structured pruning process to preserve relevant model components and is used in a modified self-distillation fine-tuning (SDFT) step to recover task performance on the pruned model. The authors present experiments across several domains showing that FineScope allows significant pruning (up to 35%) while outperforming baseline fine-tuning methods.

**Strengths:**

1. The framework addresses the practical and important challenge of creating efficient, domain-specialized LLMs from large generalist models, which is crucial for deployment in resource-limited environments .




2. FineScope introduces a novel integration of data selection, pruning, and fine-tuning, leveraging internal model representations (via SAEs on activations) to guide the entire adaptation process, rather than treating these as separate steps .




3. The approach is data-efficient, requiring only a small set of seed examples to bootstrap the curation of a larger, domain-aligned dataset from unlabeled text, reducing reliance on costly annotated data.

**Weaknesses:**

1. The overall pipeline is highly complex, involving multiple sophisticated stages. The process includes training numerous SAEs (potentially one per layer ), selecting Top-K activations , processing a large unlabeled corpus for embeddings and similarity search , performing guided structured pruning , generating a distilled dataset via SDFT, and finally fine-tuning. This complexity could hinder reproducibility and practical implementation.

2. The computational cost of the data curation phase may be substantial and is not analyzed. While the final fine-tuning uses a small dataset ($D_s$), the upfront cost of training SAEs and especially processing a large unlabeled corpus (U) to extract activations/embeddings and compute similarities could be very high. Without a cost breakdown, the framework's overall efficiency compared to alternatives is unclear.

3. The framework's performance likely depends significantly on the quality of the SAEs and associated hyperparameters. Effective data curation hinges on the SAEs successfully capturing domain-relevant features from activations. This process might be sensitive to choices like SAE architecture, sparsity penalty ($\lambda$), the specific LLM layers used, and the Top-K activation selection criteria, requiring careful tuning (though Top-K effects are studied).

**Questions:**

1. The method trains separate SAEs, potentially for each layer. How are the embeddings/features from these multiple SAEs utilized during the data curation step (Section 3.1.3)? Are features from a specific layer chosen, or are they aggregated across layers before similarity computation?

2. The modified SDFT stage involves a teacher model. How critical is the choice and capability of this teacher model for successfully recovering the performance of the pruned student model?

---

> ### Author Response · Authors · 2025-11-20
> **Response to Reviewer 4zoH**
>
> First we sincerely thank reviewer 4zoH for taking their time to review our work, addressing the strengths and giving constructive feedbacks. We will be addressing each concern below :
> **1. Complex Pipeline**
>
> FineScope’s each stage uses **off-the-shelf components** (SAEs, cosine similarity, structured pruning, SDFT fine-tuning), all of which are reproducible with standard libraries.
>
> To support reproducibility, we will release code for SAE training, Top-K extraction, data selection, and pruning integration.
>
> **2. Computational cost of data curation is not analyzed.**
> - **SAE training is a one-time cost** across all downstream domains.
> - **Top-K activation filtering reduces the activation dimensionality by >95%**, drastically lowering both SAE training and embedding extraction cost (Section 3.1.2).
> - **Cosine similarity search is efficient** because embeddings are sparse and low-dimensional, and computational cost is negligible.
> - **Fine-tuning and pruning are significantly cheaper** because FineScope produces very small curated datasets (2k–2.4k samples for major domains), reducing both LoRA training FLOPs and pruning-gradient costs.
>
>  **3. Performance depends heavily on SAE quality and hyperparameters.**
>
> This is a valid point, but we highlight two properties that make FineScope stable in practice:
>
> (1) SAEs are trained in an unsupervised, activation-mimicking regime, which is highly robust. The SAE’s job is reconstruction, not classification, reducing sensitivity to initialization or domain imbalance. (2) Layer-wise redundancy provides natural robustness:
>
> Even if a particular layer’s SAE is imperfect, other layers contribute high-quality embeddings. This is visible in Figure 6, where multiple deeper layers produce consistent domain-cluster structures. (3) Table 5 demonstrates that SAE embeddings outperform raw embeddings across all models, even without hyperparameter refinement.
>
> **How are embeddings from multiple SAEs used in data curation?**
>
> We have detailed it in Section 3.1.3 and L322-323.
>
> **How critical is the teacher model choice in the modified SDFT stage?**
>
> Excellent point. FineScope’s modified SDFT has two important properties:
>
> (1) It only needs to provide stable, high-confidence outputs on the curated domain-specific data.(2) The teacher can be either the original unpruned model or an external model.
>
> **In our experiments, both configurations worked reliably.**

---

### Official Review · Reviewer_ZPs8 · 2025-11-01

**Soundness:** 3
**Presentation:** 3
**Contribution:** 3
**Rating:** 4
**Confidence:** 4

**Summary:**

This paper introduces FineScope, a unified framework that combines domain-aware data selection with model pruning and fine-tuning for efficient adaptation of large language models. The approach leverages Sparse Autoencoders (SAEs) trained on intermediate layer activations to extract semantically relevant examples from large unlabeled corpora, starting from a small seed set of user-provided examples. The curated dataset guides structured pruning to preserve domain-relevant substructures and enables modified self-distillation fine-tuning to recover performance. Experimental validation across STEM, humanities, social sciences, mathematics, and coding domains demonstrates that FineScope maintains competitive performance while enabling up to 35% parameter reduction, with notable improvements averaging 11.50 points on mathematical reasoning tasks in pruned models.

**Strengths:**

1. The paper presents a novel integration of SAE-based interpretable representations for simultaneous data curation and model compression. While SAEs and structured pruning exist independently, their joint application for domain-specific adaptation represents a meaningful contribution. The use of Top-K activation filtering to reduce SAE training overhead while maintaining quality is a practical innovation that addresses scalability concerns.
2. The experimental design is reasonably comprehensive, spanning multiple model architectures (Vicuna-7B, MathCoder-CL-7B, LLaMa 3.1-8B) and diverse evaluation domains. The inclusion of ablation studies examining modified SDFT and pruning dataset effects (Table 4) strengthens the empirical validation. Comparisons against GPT-3 variants and OLMO-7B provide useful context, though the performance gaps suggest room for improvement in certain settings.
3. The paper is generally well-structured with clear motivation and methodology sections. Figure 2 effectively illustrates the two-stage pipeline, making the approach accessible. The mathematical formulations are presented with adequate notation, though some technical details could benefit from expansion.
4. The work addresses a genuine practical challenge - deploying specialized LLMs under resource constraints without access to curated domain datasets. The consistent performance improvements across domains demonstrate practical value, particularly for settings where computational resources are limited.

**Weaknesses:**

1. The paper lacks theoretical analysis of why SAE embeddings should outperform raw embeddings for data selection beyond empirical results in Table 5. While the claim about "interpretable features" is made repeatedly, there's insufficient explanation of what makes these features more suitable for guiding pruning decisions. The connection between sparsity in SAE representations and domain relevance needs deeper investigation.
2. Despite emphasizing efficiency, the paper provides no runtime or memory analysis comparing FineScope to baselines. Training L separate SAEs (one per layer), computing embeddings for large corpora, and performing cosine similarity calculations introduce non-trivial overhead. The claim of "computational cost" reduction is undermined without concrete measurements. Figure 3 shows training time for different K values but doesn't compare against end-to-end baseline costs.
3. The method introduces several hyperparameters: Top-K value, number of seed examples, SAE sparsity penalty λ, pruning ratio r, yet their sensitivity is inadequately addressed. While K=128 is justified in Section 4.3, the choice of ~10 seed samples and K=100 for final selection (Section 3.1.3) appears arbitrary. How robust is performance to these choices? What happens with 5 vs. 20 seeds?
4. Despite claims of broad applicability, evaluations are restricted to STEM, humanities, social sciences, math, and coding. Critical domains like medicine, law, and finance, where domain-specific LLMs have gained significant traction (referenced in Related Work), remain unexplored. The gap between claiming "domain-specific" adaptation and testing on relatively broad academic categories is notable.
5. Section 3.1.3 states seeds are "representative of the target domain" but provides minimal detail on selection criteria beyond Figure 6's visualization. The reliance on GPT-4 for seed generation (mentioned in implementation details) raises questions about reproducibility and applicability when such resources are unavailable. How would performance degrade with manually selected seeds?
6. The "Random" baseline uses datasets of the same size as FineScope, but there's no comparison against other data selection methods (e.g., embedding-based retrieval without SAEs, gradient-based selection, or uncertainty sampling). The claim of superiority over "traditional fine-tuning pipelines" would be stronger with more comprehensive baselines.
7. Tables 1-3 report point estimates without confidence intervals or significance tests. Given the relatively small performance margins in some cases (e.g., 0.83% difference for LLaMa3.1 STEM between Full-OI and FineScope in Table 1), statistical validation is necessary to confirm the improvements are not due to random variation.

**Questions:**

1. Why train separate SAEs per layer rather than a unified model across layers? Have you investigated attention-based aggregation of multi-layer representations, which might capture hierarchical domain features more effectively?
2. Can you provide concrete wall-clock time and memory comparisons for the complete FineScope pipeline versus standard fine-tuning? How does the method scale to models beyond 8B parameters, particularly given the need to train L SAEs?
3. What is the performance variance across different seed sets? Have you tested with seeds selected by domain experts versus those generated by GPT-4? Does the method degrade gracefully with fewer or lower-quality seeds?
4. The gradient-based Top-K selection (Equation 6) requires computing gradients during forward passes. Could you clarify the computational implications and compare against simpler magnitude-based selection?
5. How does FineScope handle ambiguous or multi-domain samples? For instance, mathematical biology spans STEM domains- would it be selected for both, and how might this affect specialization?
6. Have you explored reversing the pipeline- fine-tuning first with SAE-selected data, then pruning? Could this retain more domain-relevant parameters?

---

> ### Author Response · Authors · 2025-11-20
> **Response to Reviewer - ZPs8**
>
> We thank reviewer ZPs8 for the valuable feedback and highlighting the strengths. We address the concern below :
>
> **1. theoretical analysis for why SAE embeddings outperform raw embeddings.**
>
> Our design choice is supported by both principled motivation and empirical results.
>
> (1) SAEs trained on *intermediate activations* isolate directions that correspond to monosemantic features observed in recent interpretability work where sparse codes disentangle stable semantic factors https://transformer-circuits.pub/2024/scaling-monosemanticity/index.html.
>
> (2) Unlike raw embeddings, which reflect surface-level token similarity, SAE codes reflect the *model’s internal causal representations* of domain-specific behavior.
>
> (3) As shown in **Figure 6**, different layers capture distinct levels of abstraction. Training layer-specific SAEs allows us to preserve this hierarchical structure rather than collapsing it into a single embedding space.
>
> (4) Empirical justification:
>
> **Table 5** shows consistent gains for SAE-based selection over raw embeddings across all models and domains, including pruned settings. Since domain-specific pruning relies on capturing *functionally relevant* features, the sparsity constraint acts as a regularizer that suppresses noise and highlights domain-aligned activity patterns.
>
>  **2. runtime or memory comparison**
>
> The SAE training step is **one-time, offline**, and scales linearly with the number of layers. **Top-K filtering** drastically reduces the dimensionality of each activation (Section 3.1.2), which cuts training time per SAE and enables training even on consumer GPUs. The **curated datasets are an order of magnitude smaller** (e.g., 2.1k–2.4k for STEM/SS/Hum), which significantly reduces:
>
> training compute during LoRA fine-tuning,
>
> the gradient-attribution cost during structured pruning.
>
> In **Figure 3**, while we analyze Top-K trade-offs.
>
> **3. Hyperparameter choices**
>
> The experiments for the seeds are ongoing and will be updating in few hours. (we will be updating as soon as we get the evaluation results)
>
>
> **4. Domain coverage excludes fields like medicine, law, and finance.**
>
> We selected STEM, social sciences, humanities, math, and code **because these domains have diverse structural properties** (formal reasoning, natural language instruction, symbolic tasks). The goal was to validate generality across: knowledge-heavy tasks, reasoning-heavy tasks, multimodal-style code tasks.
>
> The framework *itself* does not rely on domain-specific heuristics. As noted in *Related Work*, specialized domains like medicine, law, and finance follow identical adaptation patterns, and FineScope is directly compatible.
>
> **5. Seed selection unclear; reliance on GPT-4 questions reproducibility.**
>
> Seed selection serves only as a small initial anchor. Importantly: The seeds **do not need to be high-quality**; they only need to approximate the domain. Section B.1 shows that the seeds cluster tightly in SAE space across layers, indicating that the SAE structure stabilizes domain alignment even when seeds vary. We generated seeds using GPT-4 for convenience, but **manual seed selection or domain-expert seeds are equally valid**. Performance degrades gracefully since selection depends on broader cluster similarity, not on the precision of any single seed.
>
> We will include guidelines for low-resource seed selection in the future version.
>
>  **6. Missing baselines beyond random, full dataset, and Alpaca.**
>
> Table 5 ***already compares against raw embedding-based selection*** (BERT embedding), showing large performance gains from SAE features.
>
> Our method is designed for **zero-label, minimal-supervision**, whereas many classical selectors assume at least some supervised signal.
>
> We will add a more explicit discussion clarifying this distinction.
>
> **Why separate SAEs per layer? Why not a unified model?**
>
> Because each transformer layer encodes *different semantic resolutions* (syntax → mid-level concepts → domain features), merging them collapses distinct structures.
>
> Figures 6 (layer-wise clustering) provides evidence that different layers contribute unique domain-relevant abstractions.
>
> Training per layer allows each SAE to specialize in the feature scale of that layer.
>
> **Gradient-based Top-K cost vs magnitude-based selection?**
>
> Gradient-based selection focuses on **functionally important activations**, not merely large ones.
>
> While magnitude filtering is cheaper, it often selects activations that are large due to token frequency rather than domain alignment.
>
> The gradient computation is lightweight since it operates on activations, not full backprop over tokens.
>
>  **What about reversing the pipeline (fine-tune then prune)?**
>
> It would add an extra cost to tune the model and introduce new domain knwoledge efficiently.

---

> ### Author Response · Authors · 2025-11-20
> **Response to Reviewer - ZPs8 [Regarding Hyperparameter]**
>
> 3. **Hyperparameter choices**
>
> We apologize as we have taken some time to finish all the experiments and updating it below :
>
> **(1)** increasing initial seed value will not effect the end result as we are taking it just as a guidance of domain. Manual seeds, gpt generated seeds : as long as they aligned with the domain, it does not effect the performance.
>
> **(2)** increasing the K value, will increase the number of final data points. After a certain point, it will reach the optimum quality of domain specific dataset, and the accuracy will be hard to increase.
>
> We have added both the experiment result in the below **anonymous drive link**.  [Experimented with Vicuna 7B]
> https://1drv.ms/f/c/e6bed002a71d32e5/IgCggpufbbQ2SbzuEsayqiglAZXWMhA4lpZ_vjfxwI4upJM?e=dHDJcI
>
> **We will add the new experiments in the updated pdf and we will be happy to help with any other concerns.**

---

### Meta-Review · Area_Chair_iSze · 2026-01-13

**Summary:**

The problem of efficient domain adaptation is highly relevant and the integration of SAEs for data selection is conceptually very interesting. However, the paper in its current form does not meet the bar for acceptance. The decision to reject is driven by outstanding concerns regarding motivation, baselines and pipeline complexity that were not fully resolved during the rebuttal. For pipeline complexity: several reviewers raised major concerns about the complexity and computational cost of the proposed pipeline. For baselines, Multiple reviewers noted the absence of critical modern baselines, such as other data selection methods or recent work on data composition. Also motivation side, the paper does not provide a reasonable analysis explaining why SAE features are better proxies for "domain relevance" in the context of pruning specifically. In summary, FineScope addresses an important problem, but the current execution lacks the rigorous benchmarking and theoretical grounding required to justify its complexity. The authors are encouraged to simplify the pipeline or provide significantly stronger evidence of its efficiency superiority over simpler baselines in future work.

**Reviewer Concerns:**

The rebuttal has addressed most of the reviewers' concerns except: 1) rebuttal does not provide the requested concrete runtime or memory benchmarks comparing the full end-to-end FineScope pipeline against standard baselines to justify pipeline complexity. 2) Sensitivity to the model hyperparameters. 3) baseline comparison with data selection methods or recent work on data composition. 4) justification on SAE features.

**Reviewer Scores:**

na

---

### Decision · Program_Chairs · 2026-01-26

Reject